# Immunoglobulin A Mucosal Immunity and Altered Respiratory Epithelium in Cystic Fibrosis

**DOI:** 10.3390/cells10123603

**Published:** 2021-12-20

**Authors:** Sophie Gohy, Alexandra Moeremans, Charles Pilette, Amandine Collin

**Affiliations:** 1Department of Pneumology, Cliniques Universitaires Saint-Luc, Avenue Hippocrate 2, 1200 Brussels, Belgium; alexandramoeremans5@gmail.com (A.M.); charles.pilette@uclouvain.be (C.P.); 2Pole of Pneumology, ENT and Dermatology, Institute of Experimental & Clinical Research, Université Catholique de Louvain (UCLouvain), 1200 Brussels, Belgium; Amandine.m.collin@gmail.com; 3Cystic Fibrosis Reference Center, Cliniques Universitaires Saint-Luc, 1200 Brussels, Belgium

**Keywords:** cystic fibrosis, immunoglobulin A, mucosal immunity, respiratory epithelium

## Abstract

The respiratory epithelium represents the first chemical, immune, and physical barrier against inhaled noxious materials, particularly pathogens in cystic fibrosis. Local mucus thickening, altered mucociliary clearance, and reduced pH due to CFTR protein dysfunction favor bacterial overgrowth and excessive inflammation. We aimed in this review to summarize respiratory mucosal alterations within the epithelium and current knowledge on local immunity linked to immunoglobulin A in patients with cystic fibrosis.

## 1. Introduction

Cystic fibrosis (CF), identified in 1938 by Dorothy Andersen, is a multisystemic disease and the most common lethal genetic autosomal recessive disorder in Caucasian populations [1,2,3]. CF affects around 85,000 people worldwide [4]. Incidence varies according to race and ethnicity [3], occurring in approximately 1 in 3000 live births in people of northern European ancestry, and the carrier frequency is estimated at one in 25 people [5]. The prevalence of CF increases due to the improved survival rate linked notably to highly effective modulator therapies but also to early detection of new cases thanks to improved diagnostic methods and newborn screening [3,6].

CF is caused by mutations in the CF transmembrane conductance regulator (CFTR) gene discovered in 1989. It is located in the long arm of human chromosome 7 and encodes an anion channel belonging to the ATP-binding cassette family [7,8,9,10]. CFTR is expressed at the apical surface of all epithelia (nasal mucosa, trachea, lung, exocrine pancreas, intestines, sweat glands, kidney, salivary glands, testis, uterus) as well as in serous cells in submucosal glands, ionocytes, immune cells, smooth muscle cells, neural cells, and heart cells [11,12,13,14,15,16,17,18,19,20,21,22]. In normal large and small airways, it has been recently localized more abundantly in secretory and basal cells than in ciliated cells [23]. Its wide expression explains the systemic nature of the disease. CFTR protein is a channel that regulates transport of ions and the movement of water across the epithelial barrier. It is responsible for anion transport across the plasma membrane (chloride, bicarbonate, thiocyanate, and glutathione) [24,25,26,27] and is described to regulate other ion channels and transporters such as epithelial Na+ channel (ENaC) which is down-regulated by CFTR [17]. Over 2000 mutations in the CFTR gene have been described and 382 are known to be CF-causing [28,29].

CFTR dysfunction leads to many manifestations in the organs where it is expressed but the lung disease remains the leading cause of morbidity and mortality [3] (responsible for the mortality of 66.14% of CF patients in Europe in 2018) [30]. Symptomatic treatments, progressively implemented since the 1950s, improved survival and quality of life. Targeted treatments correcting CFTR protein defects are currently available. Correctors (lumacaftor, tezacaftor, and elexacaftor) and a potentiator (ivacaftor) [31] are able to restore CFTR folding and processing and then to improve the CFTR channel opening, respectively. These drugs improve FEV1, body mass index, and quality of life, and they decrease the sweat chloride and the number of pulmonary exacerbations [32,33,34,35,36,37,38,39].

In upper airways, almost all patients will develop chronic rhinosinusitis and nasal inflammation promoting the appearance of nasal polyps [40]. CF lung disease is a muco-obstructive disorder characterized by mucus plugging, chronic neutrophilic inflammation, and recurrent infection resulting in progressive structural lung damage including bronchiectasis and destruction of lung parenchyma [41,42]. Activated neutrophils are recruited in the airways (becoming the most prominent inflammatory cell population) and discharge a large quantity of inflammatory mediators [43,44,45]. The mutant CFTR channel does not transport antioxidants to counteract neutrophil-associated oxidative stress, which contributes to the pro- and anti-inflammatory cytokines imbalance responsible for damages in CF lung tissues (bronchiectasis and bronchomalacia) and progressive lung function decline [46]. Defective immune responses (viscous mucus, defective immune cells, and antimicrobial molecules as well as decreased mucociliary clearance) result in acute and progressive chronic lung infection with opportunistic pathogens [45]. Respiratory viruses, methicillin-sensitive *Staphylococcus aureus*, and *Haemophilus influenzae* are first prevailing and are replaced with age by more harmful and resistant pathogens such as *Pseudomonas aeruginosa*, associated with a worsening of lung function and vital prognosis [47,48,49].

In CF, the respiratory epithelium is thus the target of genetic abnormality, recurrent infections (notably by *Pseudomonas aeruginosa*) and chronic inflammation and this review will focus firstly on the structure of the respiratory epithelium in normal subjects and in people with CF. Secondly, we aimed to summarize the current knowledge on local immunity related to immunoglobulin A in patients with CF.

## 2. Alterations of the CF Respiratory Epithelium

The respiratory epithelium is the first barrier against inhaled pathogens and covers the airways from the nose to the terminal respiratory bronchioles [50]. This pseudostratified epithelium is composed of four main cell types (ciliated cells, which account for over 50% of airway epithelial cells, goblet cells, basal cells, and club cells) and acts as a chemical (antimicrobial molecules, lysozyme, lactoferrin, proteases and antiproteinases, etc.), immunological, and physical barrier [20,51,52,53,54]. Recently, several subsets of the main epithelial cells based on different transcriptional signatures were described by single cell analyses (3 for ciliated and 5 for basal and secretory cells) [55]. The first subset of ciliated cells showed markers of cilia pre-assembly such as *SPAG1*, *LRRC6*, and *DNAAF1*, while the second expressed markers of mature ciliated cells (*TUBA1A* and *TUBB4B*). The third subset expressing the pro-inflammatory serum amyloid A proteins probably took part in immune response. The first subset of secretory cells is characterized by *SCGB1A1* and serpin family members expression and shared the same patterns as club cells in the bronchioles. The goblet cells per se (second subset) expressed *MUC5B* and *MUC5AC*, *AGR2*, and *SPDEF*. The third subset are thought to be progenitors for ciliated cells and showed the expression of *DNAH*, *ANKRD*, *MUC16*, and *MUC4*. Another type of secretory cells (mucinous) expressed *MUC5B*, *TFF1*, and *TFF3*. Finally, the fifth subset would be serous secretory cells expressing lysozyme and lactoferrin. For basal cells, 5 subsets are defined: subset 1 is enriched for tumor protein P63 and cytokeratins 5 and 15; subtype 2 (proliferating basal cells) for topoisomerase II alpha and the marker of proliferation Ki-67; subset 3 (basal cells transitioning to a secretory cells) for the serpin family; subset 4 showed the AP-1 family members JUN and FOS expression; and subset 5 expressed β-catenin. Their precise roles remain unknown, and they are in similar proportions between healthy controls and patients with CF [55].

The integrity of the epithelium required for its physical barrier role is maintained by apical junction complexes including tight junctions (formed by occludins, ZO-1 among other), (hemi-)desmosomes, and adherence junctions (formed by E-cadherin, α- and β-catenins) that interact with the cytoskeleton and play a role in epithelial polarization and permeability [51,52,56].

Autopsy studies revealed pulmonary changes in CF patients such as squamous metaplasia, inflammatory infiltrates, bronchiectasis, and mucopurulent plugs [57,58]. It was associated with an altered airway epithelial specification (decrease of ciliated cells and increase of goblet cells [59]), thickening of the reticular basement membrane [60], and a degradation of structural components of the extracellular matrix (elastin, collagen, glycosaminoglycans) [61]. Epithelial-to-mesenchymal transition (EMT) is a physiological process during development and repair by which a polarized epithelium de-differentiates into mesenchymal cells [62]. It involves loss of polarity markers (e.g., ZO-1 or E-cadherin), enhanced migratory capacity, and acquisition of mesenchymal features such as spindle shape, the expression of vimentin, and the secretion of extracellular matrix proteins such as fibronectin [63,64]. After damage, EMT allows mesenchymal cells to migrate on the lesion, forming a layer that can re-differentiate into a mucociliary epithelium. Upon chronic inflammation, EMT persists and may lead to remodeling [65]. Different studies showed that remodeling occurs in CF but the presence of EMT remains elusive [66]. One study demonstrated that the expression of EMT markers (fibronectin and N-cadherin) was increased in IB3 cells (a cell line derived from CF bronchial epithelium) compared with C38 control cell line [67]. TGF-β signaling promotes EMT and these features were reversible upon TGF-β inhibition in IB3 cells. In addition, TGF-β expression is increased in CF and associated with a thickening of the reticular basement membrane. The presence of EMT and altered differentiation of epithelial cells were confirmed by more recent data that aimed to characterize the abnormal phenotype in CF bronchial epithelium, showing decreased ciliated cells number, increased goblet cells number, and increased mesenchymal markers expression [68] (Figure 1, unpublished data). Interestingly, the EMT sensitivity of CFTR-expressing epithelial cells was reversed by CFTR modulators [69] while in end-stage CF patients, single-cell transcriptome confirmed altered epithelial differentiation showing fewer basal cells and more cells transitioning to ciliated and secretory cells than in non-CF controls [55].

The role of infection in CF physiopathology is important. Deficient CFTR protein favors infection by reducing bicarbonate excretion in humans and pigs (not mice) and decreasing pH of airway surface liquid [70]. It was shown in CF porcine airway epithelial cell cultures that this acidic environment impairs *Pseudomonas aeruginosa* killing [71]. Furthermore, it alters mucociliary clearance as inhibiting bicarbonate secretion in normal adult pig trachea slows mucociliary transport [72] and impaired mucociliary clearance is also associated with reduced periciliary liquid volume [73]. The CF airway tissue responds excessively and sometimes differently to infection (viral and/or bacterial) both at the protein and at the transcriptional level than control airways. It was highlighted by recent studies for *Pseudomonas aeruginosa* and human rhinovirus infections through RNA sequencing of primary airway epithelial cells from children with CF and non-CF controls [74,75]. This can contribute to the sustained inflammatory response of the CF airways notably by increased IL-8 production at baseline [76,77], but also by prolonged IL-8 mRNA levels after *Pseudomonas aeruginosa* infection. Besides elevated IL-8 levels in sputum of patients with CF, other pro-inflammatory cytokines are upregulated, such as TNF-α and IL-1β [77]. In broncho-alveolar lavage, concentrations of IL-6, IL1 β, Th1 (INF-γ), Th2 (IL-5, IL-13), and Th17 (IL-17A) cytokines were increased compared to non-CF controls [78]. In addition to this excessive inflammation and tampered bacterial killing and clearance, innate antiviral defense seems altered in CF against notably human parainfluenza 3 virus [79] or rhinovirus [75,80].

## 3. Immunoglobulin A Mucosal Immunity in the CF Respiratory Epithelium

### 3.1. IgA Structure

Immunoglobulin (Ig) A is the most abundant Ig of the respiratory mucosa (70% of Ig) [81] and is produced by B-cells located in the mucosa-associated lymphoid tissues—more particularly bronchi-associated lymphoid tissues—functioning independently from the systemic immune system [82]. Mature B-cells acquire IgA expression by undergoing both T-cell-dependent and T-cell-independent class-switch recombination [83].

Monomeric IgA is a 160 kDa protein [84] of which two subclasses exist: IgA1 and IgA2 (the latter presenting two allotypic variants, IgA2m(1) predominant in Caucasian populations and IgA2m(2) predominant in African populations). IgA2 differs from IgA1 by its 13-amino acid deletion in the hinge region making it more resistant to bacterial proteases. IgA1, the most produced subclass, represents around 90% of IgA in serum while the proportion of IgA2 is more abundant in mucosal tissues [82]. IgA produced by bone marrow plasma cells is predominantly monomeric and present in serum (88%) while the mucosal secreted form of IgA is mostly polymeric (80%, mostly dimeric) [50,82,84], thanks to the joining chain, a polypeptide necessary for binding to the polymeric immunoglobulin receptor (pIgR) [82,84]. Serum IgA concentration increases with age and varies with sex and smoking status [85].

### 3.2. IgA Regulation and Roles

pIgR, a transmembrane protein expressed on the basolateral surface of the epithelial cells, mediates the epithelial transcytosis of dimeric IgA (d-IgA) produced by plasma cells in the lamina propria underlying the epithelium [81]. With or without binding of d-IgA, pIgR is endocytosed and delivered to the apical membrane, half of pIgR being recycled to the basolateral pole in the absence of d-IgA [86,87,88]. Expression of pIgR is regulated by multiple factors such as cytokines (IFN-γ, TNF-α, IL-1, IL-4, and IL-17), viruses, bacteria, fungi, retinoic acid and hormones [89]. During the trafficking or at the apical pole, the extracellular ligand-binding region of pIgR, known as secretory component (SC), is cleaved by an unidentified leupeptin-sensitive endopeptidase and released as free form or as a component of secretory IgA (S-IgA) [90]. In the lumen, S-IgA aggregates and neutralizes pathogens and particles preventing their attachment to the epithelium through a so-called “immune exclusion” [91]. Moreover, in vitro studies suggest the ability of specific IgA to neutralize viruses, such as influenza virus or measles, during transcytosis [92,93,94,95]. Another study showed that IgA can bind a soluble antigen in the lamina propria, the immune complex being subsequently transported into the lumen through a pIgR-dependent mechanism [96]. S-IgA does not activate the classical complement pathway but could bind the IgA Fc receptor expressed by myeloid cells and promote phagocytosis, antibody-dependent cell-mediated cytotoxicity, superoxide generation, degranulation, cytokine secretion, antigen presentation, and calcium mobilization [82,92,97]. SC also plays its own part. First, it stabilizes S-IgA and protects it from proteolytic degradation, including by neutrophil elastase [81]. Second, it improves the effective function of IgA by increasing neutralization capacity against influenza virus [98]. Finally, SC regulates the proper localization of S-IgA by linking the mucus and forming an interface between the lumen and the epithelium [99]. Free SC also has its own antimicrobial properties. For instance, it decreases *Streptococcus pneumoniae* SpsA surface protein binding in the lungs [100], it binds neutrophil chemoattractant IL-8 and other chemokines, inhibiting their activity and thereby acting as an anti-inflammatory agent [101]. Massive glycosylation is critical for non-specific roles of SC and S-IgA and a change in carbohydrates is associated with defective SC [102]. Altogether, pIgR plays an important part in mucosal homeostasis through the regulation of microbiota and epithelial inflammation.

### 3.3. IgA Immunity in CF

Concerning its role in infection and inflammation, pIgR expression has already been studied in chronic respiratory diseases, such as chronic obstructive pulmonary disease [103,104,105], and in upper airway diseases, such as allergic rhinitis and chronic rhinosinusitis [106], where it was shown to be decreased in airway epithelial cells. Regarding bacterial respiratory infections, *Streptococcus pneumoniae* is able to bind SC to invade the epithelial cells in vitro [107].

In CF, several studies showed increased IgA concentrations in serum [108], more particularly with *Pseudomonas aeruginosa* chronic infection [109] and with a more severe disease [110]. This increase was also suggested in sputum or in bronchoalveolar lavage of CF patients [111], as well as increased free SC in CF sputum [112]. Specific anti-*Pseudomonas aeruginosa* IgA has also been shown to be upregulated in serum when patients suffer from chronic infection [113] or poor pulmonary function [114], but also in sputum [115] and nasal secretion [116]. However, other studies showed decreased IgA secretion in CF saliva [117] and gastric luminal perfusate [118]. In CF research on therapies, pIgR was also used to transport alpha-1 antitrypsin into the lumen of the bronchi [119]. Recently, a study using a multimodal approach (including lung tissue, sputum, serum, primary epithelial cell cultures, and CF mice) has been conducted to assess pIgR expression and the possible mechanisms responsible for pIgR dysregulation and IgA immunity in CF lung [120]. The study showed that epithelial pIgR expression, IgA production (including *Pseudomonas aeruginosa*-specific IgA), and IgA+ B-cells were upregulated in the CF lung, sputum, and serum. Increased lymphoid aggregates around the bronchi containing B-cells are described in CF [121] and represent a possible IgA source.

However, experiments in CF human bronchial epithelial cells and in F508del mice revealed a constitutive downregulation of pIgR/SC production and d-IgA transcytosis, as recapitulated by adding CFTR inhibitors in control cells. This negative CFTR-pIgR pathway involved the activation of misfolded F508del-CFTR in CF—and subsequent unfolded protein response (UPR)—indicating that endoplasmic reticulum (ER) stress and UPR pathways are key regulatory checkpoints of IgA production at mucosal surfaces. Indeed, ER stress-activating UPR was able to downregulate SC and S-IgA secretion. An in vivo model of chronic lung infection by *Pseudomonas aeruginosa*-coated microbeads showed that infection could restore pIgR expression and IgA production in the lungs of F508del mice through an IL-17 inflammatory host response [120]. *Pseudomonas aeruginosa* infection modulates the CFTR-pIgR pathway by driving a host IL-17 response that stimulates pIgR expression and further increases ER stress and UPR activation. ER stress further caused by *Pseudomonas aeruginosa* infection might become insufficiently counterbalanced by UPR activation and was shown to contribute to IgA upregulation [122]. Propension to Th17 differentiation for T-cells with consecutive increased Th17 inflammation (able to regulate pIgR mRNA) is a phenomenon present both in CF and non-CF bronchiectasis, possibly not only related to the CFTR defect [123,124]. Other cytokines upregulated in *Pseudomonas aeruginosa* infection, such as IL-1β and TNF-α, were also able to upregulate pIgR expression. This recent study might reconciliate results demonstrating a decreased concentration of SC in saliva of CF patients. It suggests that upregulation may only selectively occur in the lungs since those are the target of acquired bacterial infection [120]. Other microbes, such as human influenza A virus, a cause of exacerbation in CF, could also increase ER stress and regulate it through the expression of its nonstructural protein 1 [125,126]. However, pIgR dysregulation in the absence of lung infection was assessed only in F508del models. Therefore, IgA secretion through pIgR remains unclear in non-F508del patients and should be further addressed in rarer CFTR mutations [120]. The main features of altered respiratory epithelium and dysregulated IgA system in patients with CF are summarized in Figure 2 and Table 1.

Concerning the relation between IgA and the microbiome, the first data in the gut and in inflammatory bowel diseases showed that proinflammatory IgA-coated bacteria could drive inflammatory bowel diseases and eradicating those bacteria could be promising in the future [127]. Recently, NOD2 deficiency in Crohn’s disease has been shown to increased reverse transcytosis of IgA that dysregulates the microbiome/IgA interactions [128]. Polyreactive IgA also participates in innate reactivity to microbiota in the gut [129] and the induction of dysbiosis by IgA deficiency has been described [130].

The lung-gut axis is the possible influence of the gut microbiota on the course of the lung disease (and vice-versa) and was recently reviewed in CF by Price et al. [131]. For example, gut microbiome modifications were associated in children with exacerbations and with *Pseudomonas* colonization [132]. Considering the possible crosstalk between the IgA production in the gut and the lung dendritic cells [133], future research on S-IgA should probably focus on the connection between the lung and the gut, both affected in CF. Indeed, IgA-bound bacteria that could promote inflammation in the respiratory/intestinal mucosa could be a target of future therapies.

## 4. Gaps in Knowledge and Perspectives

Studies on patient’s lung tissues show limitations related to the end-stage status of CF patients and smoking status of controls recruited for tissue samples (lung explants and cell cultures). The accessibility to early diseased tissue remains difficult due to ethical issues. Concerning studies on expectorations, less than 10% of CF children at 6 years of age and around 40% at 10 years [134] are able to expectorate spontaneously and bacterial infection appears at very early stage but induced sputum is effective.

Different in vitro models are currently used. First, organoids and cell cultures of airway epithelial cells derived from patients in air–liquid condition show some features observed in vivo in CF (and in other respiratory diseases) but do not recapitulate the complexity of the airways. Notably, they differ in terms of cell populations (notably basal cell subtypes), compared to in vivo [55]. However, they are promising tools for patients with rare mutations to evaluate the response to CFTR modulators.

More recent studies highlight the implication of different cell types and the interplay between them. Therefore, animal models further helped to recapitulate the complexity of CF disease and to study the molecular mechanisms of the changes and IgA-related immunity acquired in the airways of CF patients, but currently mouse models mostly fail to reproduce intrinsic, CFTR-related lung disease [135]. The function/efficacy of S-IgA in CF is thus difficult to address although Marshall et al. demonstrated that SC ability to neutralize IL-8/CXCL8 is reduced, leading to neutrophilic inflammation [112], and that this reduction is caused by defective glycosylation, essential for SC antimicrobial roles [102] and proper localization of S-IgA at the interface between the lumen and the epithelium [99].

Finally, highly effective CFTR modulators have the potential to modify microbiome of patients with CF, at least for *Pseudomonas* infection [136]. In the pipeline for future research, studying the impact of CFTR modulators on pIgR/IgA system would be interesting, both in vivo and in vitro. Furthermore, the positive or negative impact of increased IgA secretion in CF bronchial lumen remains unclear. We still do not know whether it affects their microbiological status, the long-term clinical outcome, or how it is affected by CF microenvironment (impaired mucociliary clearance, proteases, etc.).

## 5. Conclusions

In conclusion, the respiratory epithelium in CF is the target of genetic abnormalities associated with an altered mucociliary clearance, increased mucus viscosity, and defective antimicrobial molecules. This will make the bed for recurrent infections (notably *Staphylococcus aureus* and *Haemophilus influenzae* in the early years of life, progressively replaced with age by more harmful and resistant pathogens such as *Pseudomonas aeruginosa*) and chronic neutrophilic inflammation. These will lead to altered differentiation with persistent EMT and altered specification characterized by increased goblet and transitional cells, decreased ciliated cells, thickening of the reticular basement membrane, increased vimentin, and fibronectin expression.

It will also drive altered function, CF being increasingly considered as a mucosal immunodeficiency syndrome with multiple defaults of innate immunity mechanisms (defective antimicrobial molecules, neutrophils, transport of antioxidants) and with altered transepithelial electric resistance due to reduced expression of junctional proteins.

Regarding IgA immunity, recent data showed for the first time that its epithelial receptor, the pIgR, is upregulated in the respiratory epithelium. IgA production is increased ex vivo in CF lung, in sputum, and in serum, mostly linked to pathogens chronic infection. This is the opposite of what is found in other chronic respiratory diseases, such as chronic obstructive pulmonary disease and asthma, and to upper airway diseases. In those diseases, the role of *Pseudomonas aeruginosa* favoring IgA secretion is not yet assessed and the potential role of other CF airway pathogens, such as *Staphylococcus aureus*, *Haemophilus influenzae*, and *Burkholderia cepacia*, in modulating S-IgA is not known and should be further studied.

## Figures and Tables

**Figure 1 cells-10-03603-f001:**
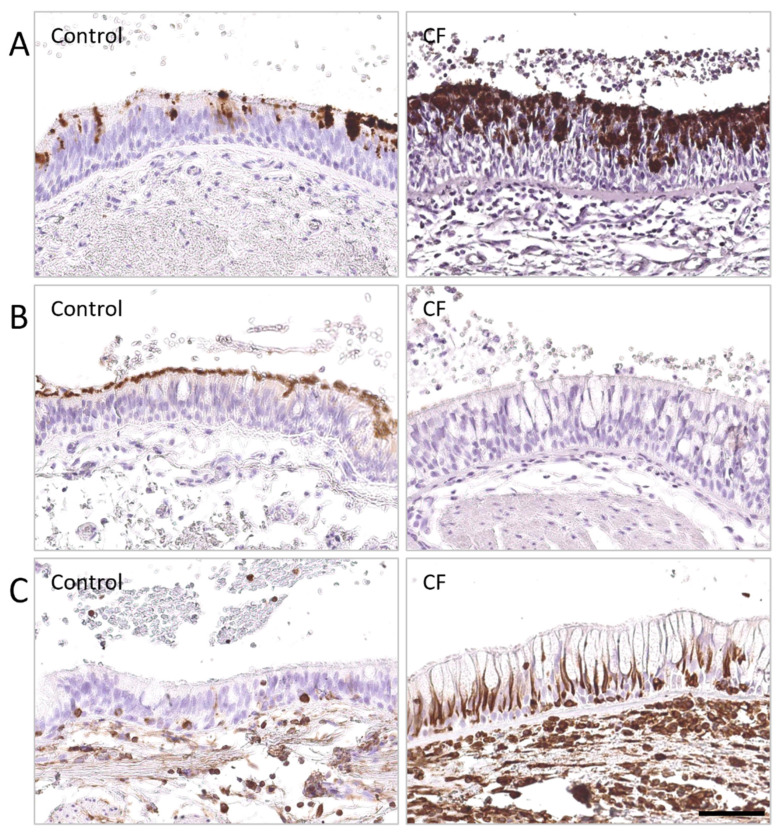
Airway epithelial differentiation is altered in the lungs of patients with severe CF. DAB stainings were performed on lung tissues from CF patients and non-CF controls, fixed in 4% formaldehyde, and embedded in paraffin wax cohort, and identified (**A**) MUC5AC (0.2 µg/mL—AM50143PU-N OriGene Technologies Inc., Rockville, MD, USA), (**B**) β-tubulin (1.16 µg/mL—T7941, Sigma-Aldrich, Saint-Louis, MO, USA), (**C**) vimentin (1 µg/mL—M0725, Dako, Carpinteria, CA, USA), respectively, and followed by detection with peroxidase-linked secondary antibody. (**A**) Increased MUCAC expression, associated with (**B**) decreased β-tubulin IV expression, were observed in the respiratory epithelium of a patient with CF, compared to a non-CF control. (**C**) Increased vimentin expression was noted in the respiratory epithelium of a patient with CF, compared to a non-CF control, as well as thickened basement membrane. Scale bar, 50 µm.

**Figure 2 cells-10-03603-f002:**
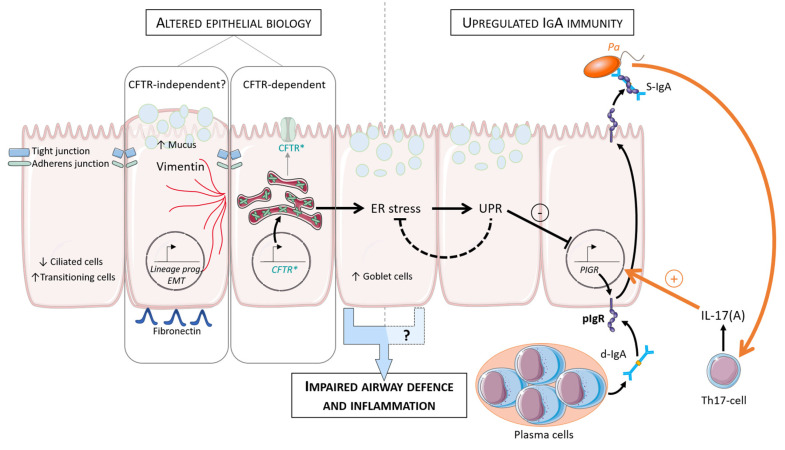
Model of epithelial alterations and IgA regulation in CF lung (adapted from [120]). Increased EMT parameters and de-differentiation of the respiratory epithelium are observed in CF (increased goblet and transitional cells and decreased ciliated cells). After ER stress and UPR activation, SC and S-IgA secretion is downregulated. *Pa* infection modulates this pathway (orange arrows). *Pa* drives host IL-17 response that stimulates *PIGR* expression and further increases ER stress and UPR activation. CF, cystic fibrosis; ER, endoplasmic reticulum; UPR, unfolded protein response; EMT, epithelial to mesenchymal transition; pIgR, polymeric immunoglobulin receptor; SC, secretory component; d-IgA, dimeric immunoglobulin; S-IgA, secretory immunoglobulin; Th, T-helper; *Pa*, *Pseudomonas aeruginosa*; prog, programming.

**Table 1 cells-10-03603-t001:** Major findings on alteration of the respiratory epithelium and the IgA/pIgR system in CF.

Findings	Model	Study
Squamous metaplasia, inflammatory infiltrates, bronchiectasis, mucopurulent plugs	Human autopsies	Bedrossian et al., 1976 [57] Sobonya and Taussig, 1986 [58]
Decrease of ciliated cells and increase of goblet cells	Lung explants	Burgel et al., 2007 [59] Collin et al., 2021 [68]
Thickening of the reticular basement membrane	Endobronchial biopsies (CF children)	Hilliard et al., 2007 [60]
Degradation of structural components of the extracellular matrix	Endobronchial biopsies (CF children)	Regamey et al., 2011 [61]
Increased expression of fibronectin and N-cadherin	IB3 cells (CF bronchial cell line)	Nyabam et al., 2016 [67]
Decreased ciliated and goblet cells numbers Increased mesenchymal markers expression Increased B-cells, peri-bronchial lymphoid aggregates	Lung explants, air–liquid interface human lower airway culture, C57BL/6N mice and F508del mice	Collin et al., 2021 [68] Frija-Masson et al., 2017 [121]
CFTR modulators reverse EMT process	Lung explants, primary human bronchial cells and cell lines (CFBE41o-)	Quaresma et al., 2020 [69]
Altered epithelial differentiation (less basal cells, more cells transitioning to ciliated and secretory cells)	Lung explants, air–liquid interface human lower airway culture	Carraro et al., 2021 [55]
Exaggerated inflammatory response (basal dysregulated production of IL-8)	Primary nasal epithelial culture	Carrabino et al., 2006 [76]
Exaggerated inflammatory response	Sputum and BAL (CF children)	Osika et al., 1999 [77] Tiringer et al., [78]
Reduced periciliary liquid volume Decreased pH of airway surface liquid impairing bacterial killing and mucociliary transport	Primary airway epithelial cultures from humans, pigs and mice Pigs trachea	Matsui et al., [73] Shah et al., 2016 [70] Pezzulo et al., 2012 [71] Birket et al., 2014 [72]
Innate antiviral defense altered in CF (human parainfluenza 3 virus)	A549 and CV-1 cells, primary bronchial epithelial culture	Zheng et al., 2003 [79]
Innate antiviral/bacterial defense altered in CF (rhinovirus, *Pseudomonas*)	Air–liquid interface human lower airway culture (CF children)	Ling et al., 2020 [75] Balloy et al., 2015 [74]
Increased IgA concentration	Serum Sputum and bronchoalveolar lavage	Hodson et al., 1988 [108] Hassan et al., 1994 [109] Van Bever et al., 1988 [110] Konstan et al., 1994 [111] Collin et al., 2020 [120]
Increased free SC	Sputum	Marshall et al., 2004 [112]
Increased specific anti-*Pseudomonas aeruginosa* IgA	Serum Sputum Nasal secretion	Schiotz et al., 1979 [113] Kronborg et al., 1992 [115] Pedersen et al., 1990 [114] Aanaes et al., 2013 [116]
Decreased IgA secretion	Saliva Gastric luminal perfusate	Oh et al., 2018 [117] Hallberg et al., 2001 [118]
*Pseudomonas aeruginosa* infection overcomes pIgR and IgA down-regulation through an IL-17 inflammatory host response	F508del mice	Collin et al., 2020 [120]
Increased Th17 inflammation	Buffy coat, F508del mice. Bronchoalveolar lavage, endobronchial biopsies	Kushwah et al., 2013 [124] Tan et al., 2011 [123]

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
