# Peer review of "Immunoglobulin A Mucosal Immunity and Altered Respiratory Epithelium in Cystic Fibrosis"

_cells, 2021, doi:10.3390/cells10123603_

Round 1

Reviewer 1 Report

Major comments:

In this review the authors sought to summarise pathophysiological changes of the respiratory mucosa in cystic fibrosis (CF) lung disease and the role of IgA in CF airway mucosal immunity. The topic is one of interest, as IgA plays an important role in mucosal defence, but its role in CF airway pathology is not well studied. Major revision, however, is required for publication.

In the manuscript’s current form, the summaries of CF epidemiology, disease, and clinical care (Subheadings 1-5) include substantial details on systemic disease, diagnosis, and therapies that ultimately detract from the focus of the manuscript, which is the respiratory epithelium. Most of the content in lines 59-75 and 98-147, for example, is not necessary. Background information on cystic fibrosis should be briefly summarised, and draw attention to the role of the airway epithelium in disease. The most novel information provided in this review is that on IgA (Subheading 6), and as a reader this is what I was most excited to learn about. Disappointingly, despite being in the title of the manuscript, findings on IgA in CF are summarised in only two paragraphs (lines 248-293). The authors highlight the role of infection, particularly with Pseudomonas aeruginosa, in upregulating S-IgA and its transcytosis mediator pIgR. They suggest a model where in the absence of infection, S-IgA production in CF airway epithelium is constitutively downregulated by endoplasmic reticulum (ER) stress and the unfolded protein response (UPR), with infection overriding this downregulation. In the conclusion of this manuscript (Subheading 7), the importance of studying IgA in CF respiratory immunity is unclear, as beyond differences in expression of IgA and pIgR in CF and other chronic respiratory conditions, there is no mention of how this advances understanding of disease, research, and potential translational impacts.

I can appreciate that compared to other aspects of airway immunity in CF, the literature on airway IgA is limited. Still, there were missed opportunities in this review to raise new questions around the role of IgA in CF lung disease and generate further interest and research in this area. This can be accomplished through restructuring and revision for a more focused and compelling review. For example, leading into the discussion of IgA, lines 165-197 currently detail epithelial barrier function and structural remodelling. This section would be more effective as subheading or series of subheadings introducing what is known on the inflammatory capacity of CF and non-CF epithelium, both at baseline and following infection, as there is ample evidence that CF airway tissue responds differently in both expression and at the transcriptional level (Zheng et al 2003, Joseph et al 2005, Carrabino et al 2006, Sutanto et al 2011, Balloy et al 2015, Ling et al 2020). This is pertinent considering the authors later highlight the role of infection in IgA upregulation. Lines 203-247 are nice summary of IgA and regulators of its expression, and would make a good subheading as an introduction to IgA itself. Following this, the findings on IgA in CF can be summarised. There are a number of topics relating to IgA, infection, and CF, that would greatly benefit this review if included as part of additional subheadings.

These include but are not limited to:

Is increased IgA and pIgR unique to adults with CF, what is known about children with milder disease, especially prior to pathogen acquisition?

Should the field be looking at the potential role of other CF airway pathogens, such as Staphylococcus aureus, Haemophilus influenzae, Burkholderia cepacia and others, in modulating S-IgA? Streptococcus pneumoniae binds to SC and pIgR, aiding in attachment and infection (Hammerschmidet et al 1997, Zhang et al 2000).

How do respiratory viruses modulate S-IgA and/or ER stress and what are potential consequences in CF? (e.g RSV – Banzhoff et al 1994, Strannegard et al 1997; Influenza – Roberson et al 2010, Mazel-Sanchez et al 2021)  What about polymicrobial infections commonly observed in CF airways?

Given the interplay of S-IgA and the microbiome in the intestinal mucosa (Nakajima et al 2018), is there a possible interplay between the airway microbiome and s-IgA in CF airways, especially in the absence of pathogens? There is evidence that IgA deficiency leads to gut dysbiosis (Catanzaro et al 2019). What is the role of the microbiome in regulating ER stress? Additionally, does IgA play a role in the gut-lung axis?

Are microbes also implicated in increased IgA observed in COPD and other inflammatory airway diseases?

Discussion about other cell types involved in secretory IgA production, as there are reports of irregularities in plasma cells, Th17 cells, and their associated signals, as a result of CFTR mutation. (Tan et al 2011, Kushwah et al 2013, Polverino et al 2019)

Is there any evidence that CFTR modulators affect or restore constitutive IgA secretion?

Do differences in cohorts, experimental design, or sample processing account for any possible differences in measured IgA across studies?

What novel techniques/experiments/models would further research into IgA in CF? Are systems biology approaches required? What are limitations of the current research?

Considering that P. aeruginosa colonisation is an issue many individuals with CF will face, how will elevated IgA affect their health long term? Does it serve as a clinical biomarker for infection? Can it be used to predict long term clinical outcomes?

I also suggest that where references in CF are lacking, that the authors consider literature on what is known on IgA in other chronic inflammatory diseases, such as Crohn’s disease (Rochereau et al 2021) and COPD (Pososukhin et al 2016). I do not expect the authors to explore all of the topics listed, but highly suggest they use some of these to generate new insights, while keeping the review focused. If the authors wish to highlight the role of infection in modulating IgA as a key takeaway, then the content overall should draw attention to the epithelium’s role in responding to infection. In summary, by condensing CF background information, focusing on the role of the epithelium in infection/inflammation, elaborating on IgA in infection/ inflammatory diseases, and connecting these to future research and potential clinical implications, this should result in a much more exciting and informative contribution to the field.

Minor comments:

If a figure is an exact copy from another publication, make sure permissions from authors and the journal are obtained and stated in the figure legend. If a figure is adapted or modified from a publication, please state clearly in the figure legend from which paper the figure is adapted.

Author Response

Response to Reviewer 1 comments

Major comments:

In this review the authors sought to summarise pathophysiological changes of the respiratory mucosa in cystic fibrosis (CF) lung disease and the role of IgA in CF airway mucosal immunity. The topic is one of interest, as IgA plays an important role in mucosal defence, but its role in CF airway pathology is not well studied. Major revision, however, is required for publication.

We thank Reviewer 1 for his/her useful comments which were carefully addressed in the revised manuscript. Please find below the point to point response.

Comment 1: In the manuscript’s current form, the summaries of CF epidemiology, disease, and clinical care (Subheadings 1-5) include substantial details on systemic disease, diagnosis, and therapies that ultimately detract from the focus of the manuscript, which is the respiratory epithelium. Most of the content in lines 59-75 and 98-147, for example, is not necessary. Background information on cystic fibrosis should be briefly summarised, and draw attention to the role of the airway epithelium in disease. The most novel information provided in this review is that on IgA (Subheading 6), and as a reader this is what I was most excited to learn about. Disappointingly, despite being in the title of the manuscript, findings on IgA in CF are summarised in only two paragraphs (lines 248-293). The authors highlight the role of infection, particularly with Pseudomonas aeruginosa, in upregulating S-IgA and its transcytosis mediator pIgR. They suggest a model where in the absence of infection, S-IgA production in CF airway epithelium is constitutively downregulated by endoplasmic reticulum (ER) stress and the unfolded protein response (UPR), with infection overriding this downregulation. In the conclusion of this manuscript (Subheading 7), the importance of studying IgA in CF respiratory immunity is unclear, as beyond differences in expression of IgA and pIgR in CF and other chronic respiratory conditions, there is no mention of how this advances understanding of disease, research, and potential translational impacts.

Response 1:

As requested by Reviewer 1,2 and 3, we adapted the manuscript by reducing previous subheading 1-5 and developing subheading 6 (as suggested in the comment 2 of the reviewer 1, in the comment 13 of reviewer 2 and in the comment 1 of the reviewer 3). The structure was changed to increase clarity: 1/ Introduction, 2/ The CF altered respiratory epithelium, 3/ Immunoglobulin A mucosal immunity in the CF respiratory epithelium and 4/ Conclusion and perspectives.

We removed thus the paragraph on hypotheses of CF pathophysiology (line 59-75) and on CF multisystemic symptoms and diagnosis (line 98-147) and made a general introduction to focus directly on respiratory epithelium and IgA in cystic fibrosis. The previous paragraph 6 was further split up in altered epithelium (+EMT) and IgA immunity with the following subheading suggested by reviewer 3:

Why respiratory epithelium is important? What is known about respiratory epithelium in CF? The missing parts are included in the conclusion and perspectives. 

Why IgA immunity is important? What is known about IgA immunity in CF? The gaps and missing parts are included in the conclusion and perspectives.

Comment 2: I can appreciate that compared to other aspects of airway immunity in CF, the literature on airway IgA is limited. Still, there were missed opportunities in this review to raise new questions around the role of IgA in CF lung disease and generate further interest and research in this area. This can be accomplished through restructuring and revision for a more focused and compelling review. For example, leading into the discussion of IgA, lines 165-197 currently detail epithelial barrier function and structural remodelling. This section would be more effective as subheading or series of subheadings introducing what is known on the inflammatory capacity of CF and non-CF epithelium, both at baseline and following infection, as there is ample evidence that CF airway tissue responds differently in both expression and at the transcriptional level (Zheng et al 2003, Joseph et al 2005, Carrabino et al 2006, Sutanto et al 2011, Balloy et al 2015, Ling et al 2020). This is pertinent considering the authors later highlight the role of infection in IgA upregulation. Lines 203-247 are nice summary of IgA and regulators of its expression, and would make a good subheading as an introduction to IgA itself. Following this, the findings on IgA in CF can be summarised. There are a number of topics relating to IgA, infection, and CF, that would greatly benefit this review if included as part of additional subheadings.

Response 2:

We revised the manuscript by adding subsections concerning the role of the respiratory epithelium during infection and we included the references (and others) suggested by the reviewer, as follows:

“The role of infection in CF physiopathology is important and the CF airway tissue responds differently to infection (viral and/or bacterial) both at the protein and at the transcriptional level than control airways as highlighted by recent studies (Balloy et al. 2015; Ling et al. 2020). This can contribute to the sustained inflammatory response of the CF airways notably by increased Il-8 production at baseline (Carrabino et al. 2006; Osika et al. 1999), but also by prolonged IL-8 mRNA levels after Pseudomonas aeruginosa infection (Joseph, Look, and Ferkol 2005). In addition, innate antiviral defense seems altered in CF against notably human parainfluenza 3 virus (Zheng et al. 2003) or rhinovirus (Sutanto et al. 2011; Ling et al. 2020).”

Comment 3:

These include but are not limited to:

A/ Is increased IgA and pIgR unique to adults with CF, what is known about children with milder disease, especially prior to pathogen acquisition?

Response 3A:

To the best of our knowledge, there is no study on pIgR/IgA expression in tissue of CF children. The accessibility to early-diseased tissue remains difficult due to ethical issues and lung transplant in CF children is extremely rare. Concerning studies on expectorations/nasal secretion, Aanaes et al. and Mauch et al. included patients above the age of 7 for the first publication and patients with a median age of 10,4 yrs for the second one, but did not performed subgroups analyses for age for secretory IgA response against Pseudomonas aeruginosa. It is known that serum IgA levels increase with age (Khan et al, JCI, 2020) in healthy population and this is now included in the revised manuscript, as follows:

“Serum IgA increases with higher age and is influenced by the sex and the smoking status (Khan et al. 2021)”

B/ Should the field be looking at the potential role of other CF airway pathogens, such as Staphylococcus aureus, Haemophilus influenzae, Burkholderia cepacia and others, in modulating S-IgA? Streptococcus pneumoniae binds to SC and pIgR, aiding in attachment and infection (Hammerschmidet et al 1997, Zhang et al 2000).

Response 3B:

Others CF pathogens mentioned play probably a role but they have not been studied so far concerning pIgR and IgA system. This gap was highlighted in the “conclusion and perspectives”.

“The potential role of other CF airway pathogens, such as Staphylococcus aureus, Haemophilus influenzae, Burkholderia cepacia and others, in modulating S-IgA is not known and should be further studied.”

As Streptococcus pneumoniae is not a relevant pathogen in CF, we include these data in the general paragraph:

“In contrast, regarding bacterial respiratory infections, Streptococcus pneumoniae is able to bind SC to invade the respiratory mucosa (Elm et al. 2004). "

C/ How do respiratory viruses modulate S-IgA and/or ER stress and what are potential consequences in CF? (e.g RSV – Banzhoff et al 1994, Strannegard et al 1997; Influenza – Roberson et al 2010, Mazel-Sanchez et al 2021)  What about polymicrobial infections commonly observed in CF airways?

Response 3C:

Polymicrobial infections do occur in CF (fungi, bacteria, mycobacteria and virus). So, as suggested, we included the articles on ER stress and influenza infection in the discussion but not those suggested on S-IgA and viruses which are only observational and not mechanistic studies more related to atopy.

“Other microbes, like human influenza A virus, a cause of exacerbation in CF, could also increase ER stress and regulate it through the expression of its nonstructural protein 1 (Roberson et al. 2012; Mazel-Sanchez et al. 2021)”

D/ Given the interplay of S-IgA and the microbiome in the intestinal mucosa (Nakajima et al 2018), is there a possible interplay between the airway microbiome and s-IgA in CF airways, especially in the absence of pathogens? There is evidence that IgA deficiency leads to gut dysbiosis (Catanzaro et al 2019). What is the role of the microbiome in regulating ER stress? Additionally, does IgA play a role in the gut-lung axis?

Response 3D:

We included a paragraph on IgA and microbiome in the gut, as follows:

“Concerning the relation between IgA and the microbiome, the first data in the gut and in inflammatory bowel diseases showed that proinflammatory IgA-coated bacteria could drive inflammatory bowel diseases and targeting those bacteria could be promising in the future (Palm et al. 2014). Recently, NOD2 deficiency in Crohn’s disease has been shown to increased reverse transcytosis of IgA that dysregulates the microbiome/IgA interactions (Rochereau et al. 2021). Polyreactive IgA also participates in innate reactivity to microbiota in the gut (Bunker et al. 2017) and the induction of dysbiosis by IgA deficiency has been described (Catanzaro et al. 2019).”

“Considering the possible crosstalk between the IgA production in the gut and the lung dendritic cells (Ruane et al. 2016), future research on S-IgA should probably focus on the connection between the lung and the gut (lung-gut axis?), both affected in CF. Indeed, IgA-bound bacteria could promote inflammation in the respiratory/intestinal mucosa and be the target of future therapies.”

This is very exciting but to date there is no data described in the lung (except an abstract on asthma: https://www.atsjournals.org/doi/abs/10.1164/ajrccm-conference.2018.197.1_MeetingAbstracts.A7704)

E/ Are microbes also implicated in increased IgA observed in COPD and other inflammatory airway diseases?

Response 3E:

IgA is decreased in all other chronic respiratory diseases (COPD, asthma, chronic rhinitis) and Pseudomonas aeruginosa is isolated from only 5-10% of COPD patients during exacerbations (Gallego, Pomares et al. 2014). COPD patients were not classified according to their bacterial colonization (including Pseudomonas aeruginosa) in studies focusing IgA immunity, excluding the possibility to study the consequences of infections (Pilette et al. Ajrccm, 2001; Polosukhin et al, Ajrccm, 2011 and Gohy et al, Ajrccm, 2014) but interestingly, viral infected airways in COPD were locally IgA-deficient in patients with COPD (Polosukhin et al. Ajrccm, 2011). Furthermore, Pseudomonas aeruginosa infection in COPD and in CF displays noticeable differences: only a small proportion of isolates are mucoid and COPD patients generally clear Pseudomonas aeruginosa, contrary to CF patients (Murphy, Brauer et al. 2008). In CF, evidence showed that a more pronounced inflammation, due to prolonged Pseudomonas aeruginosa infection (notably through IL-17), which drives an increased pIgR mediated IgA secretion in CF which is not present in COPD, including Pseudomonas aeruginosa-specific response (Collin et al, Ebiomed, 2020). In upper airway diseases, specific IgA antibodies to Staphylococcus aureus antigens was significantly decreased in nasal secretions from patients with chronic rhinitis with nasal polyps (Hupin et al. Allergy, 2013.).

This is mentioned in the revised manuscript, as follows:

“In those diseases, the role of Pseudomonas aeruginosa favoring IgA secretion is not yet assessed”.

F/ Discussion about other cell types involved in secretory IgA production, as there are reports of irregularities in plasma cells, Th17 cells, and their associated signals, as a result of CFTR mutation. (Tan et al 2011, Kushwah et al 2013, Polverino et al 2019)

Response 3F :

We added the data on Th17 cells and increased lymphoid aggregates but B-cells data were already described in the manuscript:

“Propension to Th17 differentiation for T-cells with consecutive increased Th17 inflammation (capable to regulate pIgR mRNA) is a phenomenon present both in CF and non-CF bronchiectasis, possibly not only related to CFTR defect (Tan et al. 2011; Kushwah, Gagnon, and Sweezey 2013).”

“Increased lymphoid aggregates around the bronchi containing B-cells are described in CF (Frija-Masson et al, 2017) and represent a possible IgA source.”

G/ Is there any evidence that CFTR modulators affect or restore constitutive IgA secretion?

Response 3G:

To our knowledge, no study focused on IgA in patients treated with CFTR modulators. The pipeline for future research is now highlighted in the “conclusion and perspective”.

“Highly effective CFTR modulators have the potential to modify microbiome of patients with CF, at least for Pseudomonas infection (Yi, Dalpke, and Boutin 2021). In the pipeline for future research on pIgR/IgA system, studying explants from patients treated with CFTR modulators would be interesting, even if the positive or negative impact of increased IgA in CF pathophysiology remains unclear, how it affects their health or long-term clinical outcome and how it is affected by CF microenvironment (impaired mucociliary clearance, proteases…).”

H/ Do differences in cohorts, experimental design, or sample processing account for any possible differences in measured IgA across studies?

Response 3H:

We do not think so as we performed the studies with the same experimental conditions and other groups, notably for COPD, confirmed our results. We really think that the presence of major infectious context in CF lung is capable to exceed the initial defect when infection is not present (ALI-culture or CF mice) or less present (COPD, asthma, chronic rhinitis).

I/ What novel techniques/experiments/models would further research into IgA in CF? Are systems biology approaches required? What are limitations of the current research?

Response 3I:

We included novel techniques/experiments/models in the conclusion and perspective section:

“Studies on patient’s lung tissues show limitations related to the end-stage status of CF patients and smoking status of controls recruited for tissue samples (lung explants and cell cultures). The accessibility to early-diseased tissue remains difficult due to ethical issues. Concerning studies on expectorations, less than 10% of CF children at 6 years of age and around 40% at 10 years (Sagel, Kapsner et al. 2001) are able to expectorate spontaneously and bacterial infection appears at very early stage but induced sputum is effective.

Different in vitro models are currently used. First, organoids and cell cultures of airway epithelial cells derived from patients in air-liquid condition show some features observed in vivo in CF (and in other respiratory diseases) but do not recapitulate the complexity of the airways. Notably, they differ in terms of cell populations (notably basal cell subtypes), compared to in vivo (Carraro, Langerman et al. 2021). But they are promising for patients with rare mutations to evaluate the response to CFTR modulators.

More recent studies highlight the implication of different cell types and the interplay between them. Therefore, animal models further helped to recapitulate the complexity of CF disease and to study the molecular mechanisms of the changes and IgA-related immunity acquired in the airways of CF patients, but currently mouse models mostly fail to reproduce intrinsic, CFTR-related lung disease (Wilke, Buijs-Offerman et al. 2011).”

“The function/efficay of S-IgA in CF was not addressed in the literature although Marshall et al. demonstrated that SC ability to neutralize IL-8/CXCL8 is reduced, leading to neutrophilic inflammation (Marshall et al. 2004), and that this reduction is caused by defective glycosylation, essential for SC antimicrobial roles (Corthesy 2010) and proper localization of S-IgA at the interface between the lumen and the epithelium (Phalipon et al.)”.

J/ Considering that P. aeruginosa colonisation is an issue many individuals with CF will face, how will elevated IgA affect their health long term? Does it serve as a clinical biomarker for infection? Can it be used to predict long term clinical outcomes?

Response 3J:

We still do not know if increased IgA is a positive or negative phenomenon in CF pathophysiology, how it affects their health or long-term clinical outcome. Actually, only specific IgG to Pseudomonas is used as a biomarker in clinical practice to assess the chronicity of Pseudomonas infection in parallel to the Leed’s criteria.

Comment 4: I also suggest that where references in CF are lacking, that the authors consider literature on what is known on IgA in other chronic inflammatory diseases, such as Crohn’s disease (Rochereau et al 2021) and COPD (Pososukhin et al 2016). I do not expect the authors to explore all of the topics listed, but highly suggest they use some of these to generate new insights, while keeping the review focused. If the authors wish to highlight the role of infection in modulating IgA as a key takeaway, then the content overall should draw attention to the epithelium’s role in responding to infection. In summary, by condensing CF background information, focusing on the role of the epithelium in infection/inflammation, elaborating on IgA in infection/ inflammatory diseases, and connecting these to future research and potential clinical implications, this should result in a much more exciting and informative contribution to the field.

Response 4:

We add the reference on COPD in the revised manuscript with the COPD data (ref 116) and include the results for Crohn’s disease in the IBD section:

“Recently, NOD2 deficiency in Crohn’s disease has been shown to increased reverse transcytosis of IgA that dysregulates the microbiome/IgA interactions (Rochereau et al. 2021).”

Minor comments:

If a figure is an exact copy from another publication, make sure permissions from authors and the journal are obtained and stated in the figure legend. If a figure is adapted or modified from a publication, please state clearly in the figure legend from which paper the figure is adapted.

Figure 1 is an original figure and Figure 2 was adapted from Collin A. et al. 2020, as mentioned in the revised legend.

“Model of epithelial alterations and IgA regulation in CF lung (adapted from (Collin et al. 2020)).”

Reviewer 2 Report

General comment:

The authors provide an interesting review on the role and importance of mucosal immunity, in particular IgA and possible alterations in CF airways. I have some comments and suggestions to further help improve the manuscript, which I will provide in the order that they are discussed.

Line 26:

“The improving diagnostic methods and treatments as well as the newborn screening have impacted the epidemiology of CF, its prevalence increasing due to the improved survival rate [3,6].”

This sentence reads a bit difficult. The prevalence of CF increases due to the improved survival rate which is due to the new and effective modulator therapies. Additionally, improved diagnostic methods and newborn screening increase the number of new cases detected (i.e. the incidence), thereby also increasing the total number of CF cases (new and existing), and thereby thus also affecting the prevalence. Could this sentence hence be re-formulated for clarity?

Line 36:

“nine consensus sequences accessible for phosphorylation by protein kinase A”

Please update the number of phosphorylation sites according to more recent literature, such as a review by Della Sala, 2021 (https://doi.org/10.3389/fphys.2021.690247) and an original article by the Lukacs group (Schnur, 2019; doi: 10.1038/s41598-019-48971-y)

Line 38:

“CFTR is expressed at the apical surface of all epithelia. It has been localized in the epithelial ciliated cells (nasal mucosa, trachea, lung, exocrine pancreas, intestines, sweat glands, kidney, salivary glands, testis, uterus) as well as in serous cells in submucosal glands, ionocytes, immune cells, smooth muscle cells, neural cells and heart cells [13-24].”

CFTR expression in respiratory epithelium should be updated according to recent literature: Okuda, 2021 (doi: 10.1164/rccm.202008-3198OC), Carraro, 2021 (https://doi.org/10.1038/s41591-021-01332-7). For example, the ciliated cell does not seem the most important CFTR expresser. The RNAseq data revise the historical data obtained on CFTR native expression, and this should be included in the review.

Line 64:

“The « high-salt hypothesis » would explain transepithelial potential difference”. This sentence is not clear.

You would need to add to this sentence how high salt affects TE potential difference, is it increased or decreased?

Line 65: “A defective CFTR protein prevents chloride from following sodium influx, leading to a chloride retain in the airway surface liquid (more negatively charged).”

Sentence needs rephrasing. According to the high salt hypothesis, missing or defective CFTR causes reduced transepithelial Cl- conductance and hence an elevated concentration of sodium and chloride in airway surface liquid (ASL).

Line 59: “Three hypotheses were postulated...”

It is not so clear which is the third hypothesis underlying CF pathophysiology, do the authors mean the impaired bicarbonate secretion? Can authors clarify this?

Line 121. Diagnosis of CF

A recent paper describes the use of organoid morphology (ROMA) as a novel way of diagnosing CF (Cuyx 2021, doi: 10.1136/thoraxjnl-2020-216368), with the potential for difficult to diagnose cases, such as intermediate sweat chloride. Can the authors include a short discussion of the use of this assay in specific cases for diagnosis? How do the authors believe it fits in the diagnosis pipeline?

Line 159: Targeted treatments - “Potentiator (ivacaftor) [60] and correctors (lumacaftor, tezacaftor and elexacaftor) are able to restore CFTR folding and processing and then activate CFTR.”

Can the authors change the order in which they describe the potentiators and correctors, as the correctors improve processing and trafficking, whereas the potentiator increases the open probability of the CFTR channel? Also, CFTR activation is not the same as CFTR potentiation.

Line 167: “four main cell types (ciliated cells – accounting for over 50% of airway epithelial cells –, goblet cells, basal cells and club cells)”.

Based on recent scRNAseq data there are more cell subsets than classically described. It would be nice to update this with information from the latest scRNAseq data, for example from seminal work published recently in Nat Commun (Carraro, 2021, (https://doi.org/10.1038/s41591-021-01332-7). It is also important to mention that cell composition differs between health and disease (see same paper).

Line 189: Regarding EMT discussion, it would be nice to also include a brief discussion of recent work done by the Amaral lab (Quaresma, 2020, https://doi.org/10.1038/s41419-020-03119-z). In this work, there is not so much evidence for a decrease in epithelial markers, but rather an increase in mesenchymal markers, suggesting a partial EMT phenotype in CF airway epithelium.

Figure 1: can the authors provide more experimental detail for the stainings shown? Since this is a review article without a materials and methods section, this information is currently lacking and should be added to the legend. So which antibody (cat nr, dilution), 2° ab with HRP, DAB, H&E,... Also, conclusions are now drawn in the legend stating a decrease or increase for these markers in CF compared to non-CF, but then the non-CF stainings should also be included to be complete and to support these statements.

Figure 2: it is not so clear to me what the blue vesicles are drawn within the epithelial cells. Can the authors annotate this structure?

Paragraph 6. IgA mucosal immunity in the CF altered respiratory epithelium

I would suggest to further split this up in altered epithelium (EMT) and IgA, because now it is a very long paragraph.

Regarding the IgA part, the review provided by the authors on IgA’s role in mucosal immunity in CF airways is very interesting, but what would be nice is to give the readers a clearer overview of what has been done, with which outcomes. I would therefore recommend making a table in which the main findings for each study (cell or animal model, mouse or human, read-outs regarding IgA expression, +/- infection, ...). Also, it would be nice if the authors can provide a summary at the end of this paragraph, of what they think is currently the consensus in all findings obtained so far, and which questions remain unanswered and in need of further investigation.

Author Response

Response to Reviewer 2 comments

General comment:

The authors provide an interesting review on the role and importance of mucosal immunity, in particular IgA and possible alterations in CF airways. I have some comments and suggestions to further help improve the manuscript, which I will provide in the order that they are discussed.

We thank Reviewer 2 for his/her useful comments which were carefully addressed in the revised manuscript. Please note that some sections were removed according to the reviewer 1 and 3 comments and some suggestions are not included anymore in the revised manuscript. Please find below the point to point response.

Comment 1: Line 26: “The improving diagnostic methods and treatments as well as the newborn screening have impacted the epidemiology of CF, its prevalence increasing due to the improved survival rate [3,6].” This sentence reads a bit difficult. The prevalence of CF increases due to the improved survival rate which is due to the new and effective modulator therapies. Additionally, improved diagnostic methods and newborn screening increase the number of new cases detected (i.e. the incidence), thereby also increasing the total number of CF cases (new and existing), and thereby thus also affecting the prevalence. Could this sentence hence be re-formulated for clarity?

Response 1: We modified the sentence as suggested, as follows:

“The prevalence of CF increases due to the improved survival rate linked notably to high effective modulator therapies but also to new cases early detected thanks to improved diagnostic methods and newborn screening”.

Comment 2: Line 36: “nine consensus sequences accessible for phosphorylation by protein kinase A” Please update the number of phosphorylation sites according to more recent literature, such as a review by Della Sala, 2021 (https://doi.org/10.3389/fphys.2021.690247) and an original article by the Lukacs group (Schnur, 2019; doi: 10.1038/s41598-019-48971-y)

Response 2: Because 9 sequences are formally included in R-domain and a last one in the R or the NBD1, we adapted the revised manuscript as follows:

“Ten sequences in CFTR are accessible for phosphorylation by protein kinase A, a key event for channel opening (Cant, Pollock, and Ford 2014; Rowe, Miller, and Sorscher 2005; Schnur et al. 2019).”

Comment 3: Line 38: “CFTR is expressed at the apical surface of all epithelia. It has been localized in the epithelial ciliated cells (nasal mucosa, trachea, lung, exocrine pancreas, intestines, sweat glands, kidney, salivary glands, testis, uterus) as well as in serous cells in submucosal glands, ionocytes, immune cells, smooth muscle cells, neural cells and heart cells [13-24].” CFTR expression in respiratory epithelium should be updated according to recent literature: Okuda, 2021 (doi: 10.1164/rccm.202008-3198OC), Carraro, 2021 (https://doi.org/10.1038/s41591-021-01332-7). For example, the ciliated cell does not seem the most important CFTR expresser. The RNAseq data revise the historical data obtained on CFTR native expression, and this should be included in the review.

Response 3: We included those two recent studies and revised the manuscript as follows :

“In normal large and small airways, it has been recently localized more abundantly in secretory and basal cells than in ciliated cells (Okuda et al. 2021)”.

“Interestingly, the EMT sensitivity of CFTR epithelial cells was reversed by CFTR modulators (Quaresma et al. 2020) while in end-stage CF patients, single cells transcriptome confirmed altered epithelial differentiation showing less basal cells and more transitioning cells to ciliated and secretory cells than in control donor (Carraro et al. 2021).”

Comment 4: Line 64: “The « high-salt hypothesis » would explain transepithelial potential difference”. This sentence is not clear. You would need to add to this sentence how high salt affects TE potential difference, is it increased or decreased?

Response 4: As requested by reviewer 1 and 3, we removed the paragraph on hypotheses of CF pathophysiology (line 59-75) and made a general introduction to focus directly on respiratory epithelium and IgA in cystic fibrosis.

Comment 5: Line 65: “A defective CFTR protein prevents chloride from following sodium influx, leading to a chloride retain in the airway surface liquid (more negatively charged).” Sentence needs rephrasing. According to the high salt hypothesis, missing or defective CFTR causes reduced transepithelial Cl- conductance and hence an elevated concentration of sodium and chloride in airway surface liquid (ASL).

Response 5: As requested by reviewer 1 and 3, we removed the paragraph on hypotheses of CF pathophysiology (line 59-75).

Comment 6: Line 59: “Three hypotheses were postulated...” It is not so clear which is the third hypothesis underlying CF pathophysiology, do the authors mean the impaired bicarbonate secretion? Can authors clarify this?

Response 6: As requested by reviewer 1 and 3, we removed the paragraph on hypotheses of CF pathophysiology (line 59-75).

Comment 7: Line 121. Diagnosis of CF. A recent paper describes the use of organoid morphology (ROMA) as a novel way of diagnosing CF (Cuyx 2021, doi: 10.1136/thoraxjnl-2020-216368), with the potential for difficult to diagnose cases, such as intermediate sweat chloride. Can the authors include a short discussion of the use of this assay in specific cases for diagnosis? How do the authors believe it fits in the diagnosis pipeline?

Response 7: As requested by reviewer 1 and 3, we removed the paragraph on CF multisytemic symptoms and diagnosis (line 98-147).

Comment 8: Line 159: Targeted treatments - “Potentiator (ivacaftor) [60] and correctors (lumacaftor, tezacaftor and elexacaftor) are able to restore CFTR folding and processing and then activate CFTR.” Can the authors change the order in which they describe the potentiators and correctors, as the correctors improve processing and trafficking, whereas the potentiator increases the open probability of the CFTR channel? Also, CFTR activation is not the same as CFTR potentiation.

Response 8: We made this correction as follows:

“Correctors (lumacaftor, tezacaftor and elexacaftor) and potentiator (ivacaftor) (Ramsey et al. 2011) and correctors (lumacaftor, tezacaftor and elexacaftor) are able to restore CFTR folding and processing and then activate CFTR.”

Comment 9: Line 167: “four main cell types (ciliated cells – accounting for over 50% of airway epithelial cells –, goblet cells, basal cells and club cells)”. Based on recent scRNAseq data there are more cell subsets than classically described. It would be nice to update this with information from the latest scRNAseq data, for example from seminal work published recently in Nat Commun (Carraro, 2021, (https://doi.org/10.1038/s41591-021-01332-7). It is also important to mention that cell composition differs between health and disease (see same paper).

Response 9: We included this recent study as follows:

For the epithelial cell subtypes: “Recently, different subsets of the main epithelial cells (ciliated, basal and secretory) were described but their precise roles remain unknown and they are in similar proportions between healthy controls and patients with CF (Carraro et al. 2021).”

For disease differences: “Interestingly, the EMT sensitivity of CFTR epithelial cells was reversed by CFTR modulators (Quaresma et al. 2020) while in end-stage CF patients, single cell transcriptome confirmed altered epithelial differentiation showing less basal cells and more transitioning cells to ciliated and secretory cells than in control donor (Carraro et al. 2021)”.

Comment 10: Line 189: Regarding EMT discussion, it would be nice to also include a brief discussion of recent work done by the Amaral lab (Quaresma, 2020, https://doi.org/10.1038/s41419-020-03119-z). In this work, there is not so much evidence for a decrease in epithelial markers, but rather an increase in mesenchymal markers, suggesting a partial EMT phenotype in CF airway epithelium.

Response 10: This is now included in the revised manuscript:

“Interestingly, the EMT sensitivity of CFTR epithelial cells was reversed by CFTR modulators (Quaresma et al. 2020) while in end-stage CF patients, single cell transcriptome confirmed altered epithelial differentiation showing less basal cells and more transitioning cells to ciliated and secretory cells than in control donor (Carraro et al. 2021)”.

Comment 11: Figure 1: can the authors provide more experimental detail for the stainings shown? Since this is a review article without a materials and methods section, this information is currently lacking and should be added to the legend. So which antibody (cat nr, dilution), 2° ab with HRP, DAB, H&E,... Also, conclusions are now drawn in the legend stating a decrease or increase for these markers in CF compared to non-CF, but then the non-CF stainings should also be included to be complete and to support these statements.

Response 11: Figure 1 was adapted with control images (non-CF) and more details were added to understand the stainings shown as follows:

Airway epithelial differentiation is altered in the lung from patients with severe CF. DAB stainings were performed on a lung tissues from CF patients and controls, fixed in 4% formaldehyde and embedded in paraffin wax cohort, and identified (A) MUCAC (0,2 µg/ml – AM50143PU-N OriGene Technologies Inc., Rockville, MD, USA), (B) β-tubulin (1,16 µg/ml – T7941 Sigma-Aldrich, Saint-Louis, MO, USA), (C-D) vimentin (1µg/ml – M0725, Dako, Carpinteria, CA, USA), respectively and followed by detection with peroxidase-linked secondary antibody. (A) Increased MUCAC expression, associated with (B) decreased β-tubulin IV expression, were observed in the respiratory epithelium of a patient with CF, compared to a control. (C) Thickened basement membrane was measured below the respiratory epithelium of a patient with CF, compared to a control. (D) Increased vimentin expression was noted in the respiratory epithelium of a patient with CF, compared to a control. Scale bar, 50 µm.

Comment 12: Figure 2: it is not so clear to me what the blue vesicles are drawn within the epithelial cells. Can the authors annotate this structure?

Response 12: Blue vesicles are used to identified goblet cells. The figure was better annotated to include the increased goblet and transitional cells numbers.

Comment 13: Paragraph 6. IgA mucosal immunity in the CF altered respiratory epithelium

I would suggest to further split this up in altered epithelium (EMT) and IgA, because now it is a very long paragraph.

Regarding the IgA part, the review provided by the authors on IgA’s role in mucosal immunity in CF airways is very interesting, but what would be nice is to give the readers a clearer overview of what has been done, with which outcomes. I would therefore recommend making a table in which the main findings for each study (cell or animal model, mouse or human, read-outs regarding IgA expression, +/- infection, ...). Also, it would be nice if the authors can provide a summary at the end of this paragraph, of what they think is currently the consensus in all findings obtained so far, and which questions remain unanswered and in need of further investigation.

Response 13:

As requested by Reviewer 1,2 and 3, we adapted the manuscript by reducing previous subheading 1-5 and developing subheading 6 (as suggested in the comment 2 of the reviewer 1, in the comment 13 of reviewer 2 and in the comment 1 of the reviewer 3). The structure was changed to increase clarity: 1/ Introduction, 2/ The CF altered respiratory epithelium, 3/ Immunoglobulin A mucosal immunity in the CF respiratory epithelium and 4/ Conclusion and perspectives.

We removed thus the paragraph on hypotheses of CF pathophysiology (line 59-75) and on CF multisystemic symptoms and diagnosis (line 98-147) and made a general introduction to focus directly on respiratory epithelium and IgA in cystic fibrosis. The previous paragraph 6 was further split up in altered epithelium (+EMT) and IgA immunity with the following subheading suggested by reviewer 3:

Why respiratory epithelium is important? What is known about respiratory epithelium in CF? The missing parts are included in the conclusion and perspectives. 

Why IgA immunity is important? What is known about IgA immunity in CF? The gaps and missing parts are included in the conclusion and perspectives.

Reviewer 3 Report

The authors have created a review full of information about Cystic Fibrosis disease. I personally have the feeling of not getting the point of this review. The title suggests that the review is about IgA but only one section of the entire paper is about IgA.    I also think that too many papers are cited. I do not think that 3-4 papers are needed for each sentence written by the authors.    In this review, there are too many sections off-topic. I do not think that a review of IgA should contain all those sections. May I suggest that the authors create only one introduction section containing the description of the disease and the therapies available so far (maybe adding also some pictures or scheme helping the readers to understand). After the introduction section, a dedicated section about IgA and how it is correlated to CF should be added and explained more appropriately.    The section about immunity should be described better: why immunity is so important in CF; what is known about immunity; what has been done so far and which are the missing parts?   The sections about diagnosis and therapies are not relevant for this review and I would remove them from the paper.   Section 6 is very long and confusing; it is very hard to follow and to read.

Author Response

Response to Reviewer 3 comments

We thank Reviewer 3 for his/her useful comment which was carefully addressed in the revised manuscript. Please find below the point to point response.

Comment 1:

The authors have created a review full of information about Cystic Fibrosis disease. I personally have the feeling of not getting the point of this review. The title suggests that the review is about IgA but only one section of the entire paper is about IgA.    I also think that too many papers are cited. I do not think that 3-4 papers are needed for each sentence written by the authors.    In this review, there are too many sections off-topic. I do not think that a review of IgA should contain all those sections. May I suggest that the authors create only one introduction section containing the description of the disease and the therapies available so far (maybe adding also some pictures or scheme helping the readers to understand). After the introduction section, a dedicated section about IgA and how it is correlated to CF should be added and explained more appropriately.    The section about immunity should be described better: why immunity is so important in CF; what is known about immunity; what has been done so far and which are the missing parts?   The sections about diagnosis and therapies are not relevant for this review and I would remove them from the paper.   Section 6 is very long and confusing; it is very hard to follow and to read.

Response 1:

As requested by Reviewer 1,2 and 3, we adapted the manuscript by reducing previous subheading 1-5 and developing subheading 6 (as suggested in the comment 2 of the reviewer 1, in the comment 13 of reviewer 2 and in the comment 1 of the reviewer 3). The structure was changed to increase clarity: 1/ Introduction, 2/ The CF altered respiratory epithelium, 3/ Immunoglobulin A mucosal immunity in the CF respiratory epithelium and 4/ Conclusion and perspectives.

We removed thus the paragraph on hypotheses of CF pathophysiology (line 59-75) and on CF multisystemic symptoms and diagnosis (line 98-147) and made a general introduction to focus directly on respiratory epithelium and IgA in cystic fibrosis. The previous paragraph 6 was further split up in altered epithelium (+EMT) and IgA immunity with the following subheading suggested by reviewer 3:

Why respiratory epithelium is important? What is known about respiratory epithelium in CF? The missing parts are included in the conclusion and perspectives. 

Why IgA immunity is important? What is known about IgA immunity in CF? The gaps and missing parts are included in the conclusion and perspectives.

Round 2

Reviewer 1 Report

I commend the authors for their rewrite of the manuscript. Some revision is still required before it is ready for publication, but the work overall is much improved. The manuscript needs editing for better flow of the content.

Major Comments:

1) The introduction is still too long and can be condensed further. Not every detail of CF and its pathophysiology is required, as this makes for tedious reading. Please limit background information to what is necessary to understand the rest of the manuscript, and try to summarise information more succinctly.

2) Please reword section 2 title as “Alterations of the CF Respiratory Epithelium.” Additionally, subheadings should not be phrased as questions, especially as vague statements like “Why is _____ important? Consider rephrasing as “Role of _____in _______” or “_______in CF airway immunity.”

3) I would combine subheadings 2.1 and 2.2 and delete lines 111-113. Alone 2.1 is not a very strong section.

4) Line 102. By what method were the subsets described? Did the study give any indication as to changes in cell functions?

5) Line 139 and Figure 1 legend. Is the control non-CF? I would write non-CF for clarity.

6) Figure 1A legend. Do you mean MUC5AC? I am unfamiliar with MUCAC.

7) Lines 151-157. Please expand this section as its own subheading, as epithelial responses play an important role in CF lung disease. There are many other papers detailing responses of the CF epithelium to bacteria and viruses, as well as other cytokines involved in inflammation, not just IL-8. Also talk about the science, not just the findings. How were these studies performed?

8) Lines 176-207 should fall under its own subheading summarising regulators of S-IgA.

9) Line 189. Move this sentence to the section on IgA and infection (216-235) and briefly mention that some pathogens may exploit elevated IgA to facilitate infection.

10) I suggest combining lines 208-214 and 275-279 into a single section on role of IgA in the CF gut. Also please rewrite the sentence on the gut-lung axis, and include at least one citation on this topic, in relation to CF if possible.

11) The conclusion should be a single paragraph that summarizes the review and provides a few key insights, and typically includes none or few citations. Some of the content that is currently included would fit better in brief standalone sections before the conclusion summarising in vitro studies of airway IgA, as well as clinical implications of IgA in CF, and the potential effects of modulator therapy inflammation and S-IgA production.  

Minor Comments:

1) Line 14, please use a different word from aggressions, as this is a confusing term.

2) Line 21, “Caucasian populations”

3) Line 24, please change statement to “, occurring in approximately 1 in 3000………” or something similar

4) Lines 64-66 “debate about the first-ever event happening.” Please be specific and use scientific or medical terminology. Any vague statements throughout the manuscript should be edited for clarity.

5) Lines 68-71. Please cite specific studies here, not a review article. Here is a suggested citation (Velsor et al. Antioxidant imbalance in the lungs of cystic fibrosis transmembrane conductance regulator protein mutant mice. Am J Physiol Lung Cell Mol Physiol. 2001). In general citing Review Articles within a Review Article is not good practice, please check throughout the manuscript that you avoid over citation of reviews.

6) line 92. This sentence is identical to the end of your abstract. Please rewrite.

7) Line 125. Again what do you mean by Aggression? Please edit for clarity.

8) Please edit line 174 as it is a bit awkward to read.

9) Lines 310-314. This sentence is too long and a bit confusing to read. Please split into smaller sentences.

Author Response

Response to Reviewer 1 comments

Major comments:

I commend the authors for their rewrite of the manuscript. Some revision is still required before it is ready for publication, but the work overall is much improved. The manuscript needs editing for better flow of the content.

We thank Reviewer 1 for his/her useful comment which was carefully addressed in the revised manuscript. Please find below the point to point response.

Major Comments:

Comment 1: The introduction is still too long and can be condensed further. Not every detail of CF and its pathophysiology is required, as this makes for tedious reading. Please limit background information to what is necessary to understand the rest of the manuscript, and try to summarise information more succinctly.

Response 1: As requested by reviewers 1 and 3, the introduction was further reduced and paragraph on CF respiratory epithelium and IgA expanded in the revised manuscript.

Comment 2: 2) Please reword section 2 title as “Alterations of the CF Respiratory Epithelium.” Additionally, subheadings should not be phrased as questions, especially as vague statements like “Why is _____ important? Consider rephrasing as “Role of _____in _______” or “_______in CF airway immunity.”

Response 2: We reworded the title of section 2 as requested, hoping that it will suit to reviewer 3 suggesting subheadings phrased as questions for R1.

Comment 3: I would combine subheadings 2.1 and 2.2 and delete lines 111-113. Alone 2.1 is not a very strong section.

Response 3: We deleted the line 111-113. We combined the 2.1 and 2.2 section and expanded them in the revised manuscript.

Comment 4:  Line 102. By what method were the subsets described? Did the study give any indication as to changes in cell functions?

Response 4: This is now clarified in the revised manuscript.

“Recently, several subsets of the main epithelial cells based on different transcriptional signatures were described by single cell analyses (3 for ciliated and 5 for basal and secretory cells) [66]. The first subset of ciliated cells showed markers of cilia pre-assembly like SPAG1, LRRC6 and DNAAF1 while the second expressed markers of mature ciliated cells (TUBA1A and TUBB4B). The third subset expressing the pro-inflammatory serum amyloid A proteins took probably part to immune response. The first subset of secretory cells is characterized by SCGB1A1 and serpin family members expression and shared the same patterns as club cells in the bronchioles. The goblet cells per se (second subset) expressed MUC5B and MUC5AC, AGR2 and SPDEF. The third subset are thought to be progenitors for ciliated cells and showed the expression of DNAH, ANKRD, MUC16 and MUC4. Another type of secretory cells (mucinous) expressed MUC5B, TFF1 and TFF3. Finally, the fifth subset would be serous secretory cells expressing lysozyme and lactoferrin. For basal cells, 5 subsets are defined: the subset 1 is enriched for tumor protein P63, cytokeratins 5 and 15; the subtype 2 (proliferating basal cells) for topoisomerase II alpha and the marker of proliferation Ki-67; the subset 3 (basal cells transitioning to a secretory cells) for the serpin family; the subset 4 showed the AP-1 family members JUN and FOS expression; and the subset 5 expressed β-catenin. Their precise roles remain unknown and they are in similar proportions between healthy controls and patients with CF [66].”

Comment 5: Line 139 and Figure 1 legend. Is the control non-CF? I would write non-CF for clarity.

Response 5: Yes, it is a non-CF control, we correct this in the revised manuscript.

Comment 6:  Figure 1A legend. Do you mean MUC5AC? I am unfamiliar with MUCAC.

Response 6: Indeed, it is MUC5AC and we adapted the manuscript accordingly.

Comment 7: Lines 151-157. Please expand this section as its own subheading, as epithelial responses play an important role in CF lung disease. There are many other papers detailing responses of the CF epithelium to bacteria and viruses, as well as other cytokines involved in inflammation, not just IL-8. Also talk about the science, not just the findings. How were these studies performed?

Response 7: We expanded the section as follows. Furthermore, all models used in the studies from the review are summarized in the table 1.

“The role of infection in CF physiopathology is important. Deficient CFTR protein favors infection by reducing bicarbonate excretion in humans and pigs (not mice) and decreasing pH of airway surface liquid [81]. It was shown in CF porcine airway epithelial cell cultures that this acidic environment impairs Pseudomonas aeruginosa killing [82]. Furthermore, it alters mucociliary clearance as inhibiting bicarbonate secretion in normal adult pig trachea slows mucociliary transport [83]. The CF airway tissue responds differently to infection (viral and/or bacterial) both at the protein and at the transcriptional level than control airways. It was highlighted by recent studies for Pseudomonas aeruginosa and human rhinovirus infections through RNA sequencing of primary AEC from children with CF and non-CF controls [84,85]. This can contribute to the sustained inflammatory response of the CF airways notably by increased IL-8 production at baseline [86,87], but also by prolonged IL-8 mRNA levels after Pseudomonas aeruginosa infection. Besides elevated IL-8 levels in sputum of patients with CF, other pro-inflammatory cytokines are upregulated, like TNF-α and IL-1β [87]. In broncho-alveolar lavage, concentrations of IL-6, IL1 β, Th1 (INF-γ), Th2 (IL-5, IL-13), Th17 (IL-17A) were increased compared to non-CF controls [88]. In addition to this excessive inflammation and tampered bacterial killing and clearance, innate antiviral defense seems altered in CF against notably human parainfluenza 3 virus [89] or rhinovirus [85,90].”

Comment 8: Lines 176-207 should fall under its own subheading summarising regulators of S-IgA.

Response 8: we adapted the revised manuscript accordingly and made more subsections as requested: a) IgA structure; b) IgA regulation and roles; c) IgA immunity in CF.

Comment 9: Line 189. Move this sentence to the section on IgA and infection (216-235) and briefly mention that some pathogens may exploit elevated IgA to facilitate infection.

Response 9:.We adapted the revised manuscript accordingly.

Comment 10: I suggest combining lines 208-214 and 275-279 into a single section on role of IgA in the CF gut. Also please rewrite the sentence on the gut-lung axis, and include at least one citation on this topic, in relation to CF if possible.

Response 10: We combined the 2 paragraphs and rewrote the sentence and included more references on the field:

“Concerning the relation between IgA and the microbiome, the first data in the gut and in inflammatory bowel diseases showed that proinflammatory IgA-coated bacteria could drive inflammatory bowel diseases and eradicating those bacteria could be promising in the future [129]. Recently, NOD2 deficiency in Crohn’s disease has been shown to increased reverse transcytosis of IgA that dysregulates the microbiome/IgA interactions [130]. Polyreactive IgA also participates in innate reactivity to microbiota in the gut [131] and the induction of dysbiosis by IgA deficiency has been described [132].

The lung-gut axis is the possible influence of the gut microbiota on the course of the lung disease (and vice-versa) and was recently reviewed in CF by Price et al. [133]. For example, gut microbiome modifications were associated in children with exacerbations and with Pseudomonas colonization [134]. Considering the possible crosstalk between the IgA production in the gut and the lung dendritic cells [135], future research on S-IgA should probably focus on the connection between the lung and the gut, both affected in CF. Indeed, IgA-bound bacteria that could promote inflammation in the respiratory/intestinal mucosa could be a target of future therapies.”

Comment 11: The conclusion should be a single paragraph that summarizes the review and provides a few key insights, and typically includes none or few citations. Some of the content that is currently included would fit better in brief standalone sections before the conclusion summarising in vitro studies of airway IgA, as well as clinical implications of IgA in CF, and the potential effects of modulator therapy inflammation and S-IgA production.  

Response 11: We added a paragraph on “Gaps in knowledge and perspectives” as suggested and reduced the conclusion (without any references) as follows:

“In conclusion, the respiratory epithelium in CF is the target of genetic abnormalities associated with an altered mucociliary clearance, increased mucus viscosity and defective antimicrobial molecules. This will make the bed for recurrent infections (notably Staphylococcus aureus and Haemophilus influenzae in the early years of life, progressively replaced with age by more harmful and resistant pathogens such as Pseudomonas aeruginosa) and chronic neutrophilic inflammation. These will lead to altered differentiation with persistent EMT and altered specification characterized by increased goblet and transitional cells, decreased ciliated cells, thickening of the reticular basement membrane, increased vimentin and fibronectin expression.

It will also drive to altered function, CF being increasingly considered as a mucosal immunodeficiency syndrome with multiple defaults of innate immunity mechanisms (defective antimicrobial molecules, neutrophils, transport of antioxidants) and with altered transepithelial electric resistance due to reduced expression of junctional proteins.”

Regarding IgA immunity, recent data showed for the first time that its epithelial receptor, the pIgR, is upregulated in the respiratory epithelium. IgA production is increased ex vivo in CF lung, in sputum and in serum, mostly linked to pathogens chronic infection. This is the opposite of what is found in other chronic respiratory diseases, such as chronic obstructive pulmonary disease and asthma, and to upper airway diseases. In those diseases, the role of Pseudomonas aeruginosa favoring IgA secretion is not yet assessed and the potential role of other CF airway pathogens, such as Staphylococcus aureus, Haemophilus influenzae, Burkholderia cepacia, in modulating S-IgA is not known and should be further studied.”

Minor Comments:

Comment 1:  Line 14, please use a different word from aggressions, as this is a confusing term.

Response 1: This was replaced byinhaled noxious materials”

Comment 2:  Line 21, “Caucasian populations”

Response 2: This was adapted in the revised manuscript.

Comment 3: Line 24, please change statement to “, occurring in approximately 1 in 3000………” or something similar

Response 3: This was adapted in the revised manuscript.

Comment 4: Lines 64-66 “debate about the first-ever event happening.” Please be specific and use scientific or medical terminology. Any vague statements throughout the manuscript should be edited for clarity.

Response 4: This sentence was removed when reducing the introduction.

Comment 5: Lines 68-71. Please cite specific studies here, not a review article. Here is a suggested citation (Velsor et al. Antioxidant imbalance in the lungs of cystic fibrosis transmembrane conductance regulator protein mutant mice. Am J Physiol Lung Cell Mol Physiol. 2001). In general citing Review Articles within a Review Article is not good practice, please check throughout the manuscript that you avoid over citation of reviews.

Response 5: We adapted the reference. We agree with the reviewer and we kept reviews only in the introduction or sometimes because the field is out of the scope of our review but the subject may be of interest for the readership (for example, Price et al.).

Comment 6: line 92. This sentence is identical to the end of your abstract. Please rewrite.

Response 6: This was adapted in the revised manuscript as follows:

“In CF, the respiratory epithelium is thus the target of genetic abnormality, recurrent infections (notably by Pseudomonas aeruginosa) and chronic inflammation and this review will focus firstly on the structure of the respiratory epithelium in normal subject and in people with CF. Secondly, we aimed to summarize the current knowledge on local immunity related to immunoglobulin A in the patients with CF.”

Comment 7: Line 125. Again what do you mean by Aggression? Please edit for clarity.

Response 7: This was replaced by: “After damage,”

Comment 8: Please edit line 174 as it is a bit awkward to read.

Response 8: This was adapted as follows: Serum IgA concentration increases with age and varies with sex and smoking status [91].”

Comment 9: Lines 310-314. This sentence is too long and a bit confusing to read. Please split into smaller sentences.

Response 9: we adapted the manuscript accordingly:

“Regarding IgA immunity, recent data showed for the first time that its epithelial receptor, the pIgR, is upregulated in the respiratory epithelium. IgA production is increased ex vivo in CF lung, in sputum and in serum, mostly linked to pathogens chronic infection. This is the opposite of what is found in other chronic respiratory diseases, such as chronic obstructive pulmonary disease and asthma, and to upper airway diseases.”

Reviewer 2 Report

I thank the reviewers for thoroughly taking into consideration the comments given in the first round of revision. I have a few more remaining comments:

Line 89: potentiators don’t activate CFTR (in contrast to cAMP agonists such as forskolin). Rather, potentiators promote the open state of the CFTR channel and only in combination with CFTR activation (forskolin) does the channel conduct anions.

Can the authors thus change this to “improve CFTR channel opening”

Line 134: “showing decreased ciliated and goblet cells numbers and increased mesenchymal markers expression [89] (Figure 1, unpublished data).”

I believe this should be “decreased ciliated and increased goblet cells...” as they show in figure 1.

Lines 319-329: The last two sentences concluding the review are both rather long and difficult to understand. Can the authors try to rephrase them for easier understanding, since here they present their final conclusions and considerations for future research?

Is it feasible, do the authors believe, to be able to get access to explant lung tissue from CF patients treated with modulators, since they are believed to not need a future lung transplantation due to improved (lung) health?

Author Response

Response to Reviewer 2 comments

General comment:

I thank the reviewers for thoroughly taking into consideration the comments given in the first round of revision. I have a few more remaining comments:

We thank Reviewer 2 for his/her useful comment which was carefully addressed in the revised manuscript. Please find below the point to point response.

Comment 1: Line 89: potentiators don’t activate CFTR (in contrast to cAMP agonists such as forskolin). Rather, potentiators promote the open state of the CFTR channel and only in combination with CFTR activation (forskolin) does the channel conduct anions. Can the authors thus change this to “improve CFTR channel opening”

Response 1: We adapted the manuscript accordingly.

Comment 2: Line 134: “showing decreased ciliated and goblet cells numbers and increased mesenchymal markers expression [89] (Figure 1, unpublished data).” I believe this should be “decreased ciliated and increased goblet cells...” as they show in figure 1.

Response 2: We adapted the manuscript accordingly.

Comment 3: Lines 319-329: The last two sentences concluding the review are both rather long and difficult to understand. Can the authors try to rephrase them for easier understanding, since here they present their final conclusions and considerations for future research?

Response 3: We adapted as follows:

“Finally, highly effective CFTR modulators have the potential to modify microbiome of patients with CF, at least for Pseudomonas infection [138]. In the pipeline for future research, studying the impact of CFTR modulators on pIgR/IgA system would be interesting, both in vivo and in vitro. Furthermore, the positive or negative impact of increased IgA secretion in CF bronchial lumen remains unclear. We still do not know whether it affects their microbiological status, the long-term clinical outcome or how it is affected by CF microenvironment (impaired mucociliary clearance, proteases…).”

Comment 4: Is it feasible, do the authors believe, to be able to get access to explant lung tissue from CF patients treated with modulators, since they are believed to not need a future lung transplantation due to improved (lung) health?

Response 4: Unfortunately, some lung transplants of patients under CFTR modulators (notably with ivacaftor) do occur because not all patients are responders to the medication or present some complications that require a lung transplant (hemoptysis; pneumothorax ect). So, it is not a frequent situation but this could happen and having access to the lung tissue could help to highlight ex vivo the effect of modulators on IgA immunity.

Reviewer 3 Report

The present version of the paper is an improved version of the review. 

The authors have re-organized the entire manuscript following the reviewer's suggestions. 

Comment 1:

I personally think that the introduction section is still quite long. In particular, I don't understand the final remark of the introduction section. You spoke about CF, its symtoms, the drugs used for CF therapy and suddenly you spoke about the aim of the review. I suggest that the authors find a better way to describe the aim of this review. 

Comment 2:

In my opinion, section 2 should be expanded: the altered epithelium should be one main focus of this review.

Minor comments: 

Line 34: there is a typo. Ten sequences...

Author Response

Response to Reviewer 3 comments

The present version of the paper is an improved version of the review. 

The authors have re-organized the entire manuscript following the reviewer's suggestions. 

We thank Reviewer 3 for his/her useful comment which was carefully addressed in the revised manuscript. Please find below the point to point response.

Comment 1:

I personally think that the introduction section is still quite long. In particular, I don't understand the final remark of the introduction section. You spoke about CF, its symtoms, the drugs used for CF therapy and suddenly you spoke about the aim of the review. I suggest that the authors find a better way to describe the aim of this review. 

Comment 2:

In my opinion, section 2 should be expanded: the altered epithelium should be one main focus of this review.

Response 1 and 2: As requested by reviewers 1 and 3, the introduction was further reduced and paragraph on CF respiratory epithelium expand (both 2.1 and 2.2, now combined). The aim was adapted as follows:

“In CF, the respiratory epithelium is thus the target of genetic abnormality, recurrent infections (notably by Pseudomonas aeruginosa) and chronic inflammation and this review will focus firstly on the structure of the respiratory epithelium in normal subject and in people with CF. Secondly, we aimed to summarize the current knowledge on local immunity related to immunoglobulin A in the patients with CF.”

Comment 3: Minor comments: 

Line 34: there is a typo. Ten sequences...

Response 3: We removed this sentence in the revised manuscript.